# Approximating Information Measures for Fields

**DOI:** 10.3390/e22010079

**Published:** 2020-01-09

**Authors:** Łukasz Dębowski

**Affiliations:** Institute of Computer Science, Polish Academy of Sciences, ul. Jana Kazimierza 5, 01-248 Warszawa, Poland; ldebowsk@ipipan.waw.pl; Tel.: +48-22-3800-553

**Keywords:** Shannon information measures, fields, invariance of completion, chain rule, 94A17

## Abstract

We supply corrected proofs of the invariance of completion and the chain rule for the Shannon information measures of arbitrary fields, as stated by Dębowski in 2009. Our corrected proofs rest on a number of auxiliary approximation results for Shannon information measures, which may be of an independent interest. As also discussed briefly in this article, the generalized calculus of Shannon information measures for fields, including the invariance of completion and the chain rule, is useful in particular for studying the ergodic decomposition of stationary processes and its links with statistical modeling of natural language.

## 1. Introduction

As it was noticed by Dębowski [1,2,3], a generalized calculus of Shannon information measures for arbitrary fields—initiated by Gelfand et al. [4] and later developed by Dobrushin [5], Pinsker [6], and Wyner [7]—is useful in particular for studying the ergodic decomposition of stationary processes and its links with statistical modeling of natural language. Fulfilling this need, Dębowski [1] has developed the calculus of Shannon information measures for arbitrary fields, relaxing the requirement of regular conditional probability, assumed implicitly by Dobrushin [5] and Pinsker [6]. He has done it unaware of the classical paper by Wyner [7], which pursued exactly the same idea, with some differences due to an independent interest.

Compared to exposition [7], the added value of the paper [1] was considering continuity and invariance of Shannon information measures with respect to completion of fields. Unfortunately, the proof of Theorem 2 in [1] establishing this invariance and the generalized chain rule contains some mistakes and gaps, which we have discovered recently. For this reason, in this article, we would like to provide a correction and a few new auxiliary results which may be of an independent interest. In this way, we will complete the full generalization of Shannon information measures and their properties, which was developed step-by-step by Gelfand et al. [4], Dobrushin [5], Pinsker [6], Wyner [7], and Dębowski [1]. By the way, we will also rediscuss the linguistic motivations of our results.

The preliminaries are as follows. Fix a probability space (Ω,J,P). Fields are set algebras closed under finite Boolean operations, whereas σ-fields are assumed to be closed also under countable unions and products. A field is called finite if it has finitely many elements. A finite partition is a finite collection of events ${\left\{B_{j}\right\}}_{j=1}^{J}\subset J$ which are disjoint and whose union equals Ω. The definition proposed by Wyner [7] and Dębowski [1] independently reads as follows:

**Definition** **1.**
*For finite partitions $\alpha ={\left\{A_{i}\right\}}_{i=1}^{I}$ and $\beta ={\left\{B_{j}\right\}}_{j=1}^{J}$ and a probability measure P, the entropy and mutual information are defined as*
(1)$$\begin{array}{cccc} H_{P}\left(\alpha \right) & :=\sum_{i=1}^{I}P\left(A_{i}\right)\log\frac{1}{P(A_{i})},\; & I_{P}\left(\alpha ;\beta \right) & :=\sum_{i=1}^{I}\sum_{j=1}^{J}P\left(A_{i}\cap B_{j}\right)\log\frac{P(A_{i}\cap B_{j})}{P\left(A_{i}\right)P\left(B_{j}\right)}. \end{array}$$
*Subsequently, for an arbitrary field C and finite partitions α and β, we define the pointwise conditional entropy and mutual information as*
(2)$$\begin{array}{cccc} H_{P}\left(\alpha ||C\right) & :=H_{P(\cdot |C)}\left(\alpha \right),\; & I_{P}\left(\alpha ;\beta ||C\right) & :=I_{P(\cdot |C)}\left(\alpha ;\beta \right), \end{array}$$
*where P(E|C) is the conditional probability of event E∈J with respect to the smallest complete σ-field containing C. Subsequently, for arbitrary fields A, B, and C, the (average) conditional entropy and mutual information are defined as*
(3)$$\begin{array}{cccc} H_{P}\left(A|C\right) & :=\underset{\alpha \subset A}{\sup}E{\;}_{P}H_{P}\left(\alpha ||C\right),\; & I_{P}\left(A;B|C\right) & :=\underset{\alpha \subset A,\beta \subset B}{\sup}E{\;}_{P}I(\alpha ;\beta ||C), \end{array}$$
*where the supremum is taken over all finite subpartitions and $E{\;}_{P}X:=\int XdP$ is the expectation. Finally, we define the unconditional entropy $H_{P}\left(A\right):=H_{P}\left(A|\left\{\emptyset ,\Omega \right\}\right)$ and mutual information $I_{P}\left(A;B\right):=I_{P}\left(A;B|\left\{\emptyset ,\Omega \right\}\right)$, as it is generally done in information theory. When the probability measure P is clear from the context, we omit subscript P from all above notations.*


Although the above measures, called Shannon information measures, have usually been discussed for σ-fields, the defining equations (Equation 3) also make sense for fields. We observe a number of identities, such as H(A)=I(A;A) and H(A|C)=I(A;A|C). It is important to stress that Definition 1, in contrast to the earlier expositions by Dobrushin [5] and Pinsker [6], is simpler—as it applies one Radon–Nikodym derivative less—and does not require regular conditional probability, i.e., it does not demand that conditional distribution ${\left(P\left(E|C\right)\right)}_{E\in J}$ be a probability measure almost surely. In fact, the expressions on the right-hand sides of the equations in (Equation 3) are defined for all A, B, and C. No problems arise when conditional probability is not regular since conditional distribution ${\left(P\left(E|C\right)\right)}_{E\in E}$ restricted to a finite field E is a probability measure almost surely [8] (Theorem 33.2).

We should admit that in the context of statistical language modeling, the respective probability space is countably generated so regular conditional probability is guaranteed to exist. Thus, for linguistic applications, one might think that expositions [5,6] are sufficient, although for a didactic reason, the approaches proposed by Wyner [7] and Dębowski [1] lead to a simpler and more general calculus of Shannon information measures. Yet, there is a more important reason for Definition 1. Namely, to discuss the ergodic decomposition of entropy rate and excess entropy—some highly relevant results for statistical language modeling, developed in [1] and to be briefly recalled in Section 3—we need the invariance of Shannon information measures with respect to completion of fields. But within the framework of Dobrushin [5] and Pinsker [6], such invariance of completion does not hold for strongly nonergodic processes, which seem to arise quite naturally in statistical modeling of natural language [1,2,3]. Thus, the approach proposed by Wyner [7] and Dębowski [1] is in fact indispensable.

Thus, let us inspect the problem of invariance of Shannon information measures with respect to completion of fields. A σ-field is called complete, with respect to a given probability measure *P*, if it contains all sets of outer *P*-measure 0. Let σ(A) denote the intersection of all complete σ-fields containing class A, i.e., σ(A) is the completion of the generated σ-field. Let A∧B denote the intersection of all fields that contain A and B. Assuming Definition 1, the following statement has been claimed true by Dębowski [1] (Theorem 2):
**Theorem** **1.***Let A, B, C, and D be subfields of J.**1.* I(A;B|C)=I(A;σ(B)|C)=I(A;B|σ(C)) (invariance of completion);*2.* I(A;B∧C|D)=I(A;B|D)+I(A;C|B∧D) (chain rule).

The property stated in Theorem 1.1 will be referred to as the invariance of completion. It was not discussed by Wyner [7]. The property stated in Theorem 1.2 is usually referred to as the chain rule or the polymatroid identity. It was proved independently by Wyner [7].

As we have mentioned, the invariance of completion is crucial to prove the ergodic decomposition of the entropy rate and excess entropy of stationary processes. But the proof of the invariance of completion given by Dębowski [1] contains a mistake in the order of quantifiers, and the respective proof of the chain rule is too laconic and contains a gap. For this reason, we would like to supplement the corrected proofs in this article. As we have mentioned, the chain rule was proved by Wyner [7], using an approximation result by Dobrushin [5] and Pinsker [6]. For completeness, we would like to provide a different proof of this approximation result—which follows easily from the invariance of completion—and to supply proofs of both parts of Theorem 1.

The corrected proofs of Theorem 1, to be presented in Section 2, are much longer than the original proofs by Dębowski [1]. In particular, for the sake of proving Theorem 1, we will discuss a few other approximation results, which seem to be of an independent interest. To provide more context for our statements, in Section 3, we will also recall the ergodic decomposition of excess entropy and its application to statistical language modeling.

## 2. Proofs

Let us write $B_{n}↑B$ for a sequence ${\left(B_{n}\right)}_{n\in N}$ of fields such that $B_{1}\subset B_{2}\subset \cdots \subset B={⋃}_{n\in N}B_{n}$. (B need not be a σ-field.) Our proof of Theorem 1 will rest on a few approximation results and this statement by Dębowski [1] (Theorem 1):
**Theorem** **2.***Let A, B, $B_{n}$, and C be subfields of J.**1.* I(A;B|C)=I(B;A|C);*2.* I(A;B|C)≥0 with the equality if and only if P(A∩B|C)=P(A|C)P(B|C) almost surely for all A∈A and B∈B;*3.* I(A;B|C)≤min(H(A|C),H(B|C));*4.* $I\left(A;B_{1}|C\right)\leq I\left(A;B_{2}|C\right)$ if $B_{1}\subset B_{2}$;*5.* $I\left(A;B_{n}|C\right)↑I\left(A;B|C\right)$ for $B_{n}↑B$.

Let $A^{c}=\Omega \backslash A$. Subsequently, let us denote the symmetric difference
(4)$$\begin{array}{c} A▵B:=(A\backslash B)\cup (B\backslash A)=(A\cup B)\backslash (A\cap B). \end{array}$$
Symmetric difference satisfies the following identities, which will be used:(5)$$\begin{array}{cc} A^{c}▵B^{c} & =A▵B, \end{array}$$
(6)$$\begin{array}{cc} A▵B & \subset (A▵C)\cup (C▵B), \end{array}$$(7)$$\begin{array}{cc} (A\backslash C)▵B & \subset (A▵B)\cup (C\cap B), \end{array}$$
(8)$$\begin{array}{cc} \left(\underset{i\in C}{⋃}A_{i}\right)▵\left(\underset{i\in C}{⋃}B_{i}\right)\; & \subset \underset{i\in C}{⋃}\left(A_{i}▵B_{i}\right). \end{array}$$
Moreover, we will apply the Bonferroni inequalities
(9)$$\begin{array}{c} 0\leq \sum_{1\leq i\leq n}P\left(A_{i}\right)-P\left(\underset{1\leq i\leq n}{⋃}A_{i}\right)\leq \sum_{1\leq i<j\leq n}P\left(A_{i}\cap A_{j}\right) \end{array}$$
and inequality P(A)≤P(B)+P(A▵B).

In the following, we will derive the necessary approximation results. Our point of departure is the following folklore fact.

**Theorem** **3**(approximation of σ-fields)**.**
*For any field K and any event G∈σ(K), there is a sequence of events $K_{1},K_{2},\cdots \in K$ such that*
(10)$$\begin{array}{c} \lim_{n\to \infty }P\left(G▵K_{n}\right)=0. \end{array}$$

**Proof.** Denote the class of sets *G* that satisfy (Equation 10) as G. It is sufficient to show that G is a complete σ-field that contains the field K. Clearly, all G∈K satisfy (Equation 10) so G⊃K. Now, we verify the conditions for G to be a σ-field.
We have Ω∈K. Hence, Ω∈G.For A∈G, consider $K_{1},K_{2},\cdots \in K$ such that $\lim_{n\to \infty }P\left(A▵K_{n}\right)=0$. Then, $A▵K_{n}=A^{c}▵K_{n}^{c}$, where $K_{1}^{c},K_{2}^{c},\cdots \in K$. Hence, $A^{c}\in G$.For $A_{1},A_{2},\cdots \in G$, consider events $K_{i}^{n}\in K$ such that $P\left(A_{i}▵K_{i}^{n}\right)\leq 2^{-n}$. Then,
(11)$$\begin{array}{c} P\left(\left(\overset{n}{\underset{i=1}{⋂}}A_{i}\right)▵\left(\overset{n}{\underset{i=1}{⋂}}K_{i}^{i+n}\right)\right)\leq \sum_{i=1}^{n}P\left(A_{i}▵K_{i}^{i+n}\right)\leq 2^{-n}. \end{array}$$
Moreover,
(12)$$\begin{array}{c} P\left(\left(\overset{\infty }{\underset{i=1}{⋂}}A_{i}\right)▵\left(\overset{n}{\underset{i=1}{⋂}}A_{i}\right)\right)=P\left(\overset{n}{\underset{i=1}{⋂}}A_{i}\right)-P\left(\overset{\infty }{\underset{i=1}{⋂}}A_{i}\right). \end{array}$$
Hence,
(13)$$\begin{array}{cc} \; & P\left(\left(\overset{\infty }{\underset{i=1}{⋂}}A_{i}\right)▵\left(\overset{n}{\underset{i=1}{⋂}}K_{i}^{i+n}\right)\right)\\ \; & \leq P\left(\left(\overset{\infty }{\underset{i=1}{⋂}}A_{i}\right)▵\left(\overset{n}{\underset{i=1}{⋂}}A_{i}\right)\right)+P\left(\left(\overset{n}{\underset{i=1}{⋂}}A_{i}\right)▵\left(\overset{n}{\underset{i=1}{⋂}}K_{i}^{i+n}\right)\right)\\ \; & \leq P\left(\overset{n}{\underset{i=1}{⋂}}A_{i}\right)-P\left(\overset{\infty }{\underset{i=1}{⋂}}A_{i}\right)-2^{-n}, \end{array}$$
which tends to 0 for *n* going to infinity. Since ${⋂}_{i=1}^{n}K_{i}^{i+n}\in K$, we thus obtain that ${⋂}_{i=1}^{\infty }A_{i}\in G$.
Completeness of σ-field G is straightforward since, for any A∈G and $P(A▵A^{\prime })=0$, we obtain $A^{\prime }\in G$ using the same sequence of approximating events in field K as for event *A*.  □

The second approximation result is the following bound:
**Theorem** **4**(continuity of entropy)**.**
*Fix an $\epsilon \in (0,e^{-1}]$ and a field C. For finite partitions $\alpha ={\left\{A_{i}\right\}}_{i=1}^{I}$ and $\alpha^{\prime }={\left\{A_{i}^{\prime }\right\}}_{i=1}^{I}$ such that $P(A_{i}▵A_{i}^{\prime })\leq \epsilon$ for all $i\in \left\{1,\cdots ,I\right\}$, we have*
(14)$$\begin{array}{c} |H\left(\alpha |C\right)-H\left(\alpha^{\prime }|C\right)|\leq I\sqrt{\epsilon }\log\frac{I}{\sqrt{\epsilon }}. \end{array}$$
**Proof.** We have the expectation $\int P\left(A_{i}▵A_{i}^{\prime }|C\right)dP=P\left(A_{i}▵A_{i}^{\prime }\right)\leq \epsilon$. Hence, by the Markov inequality we obtain
(15)$$\begin{array}{c} P\left(P\left(A_{i}▵A_{i}^{\prime }|C\right)\geq \sqrt{\epsilon }\right)\leq \sqrt{\epsilon }. \end{array}$$
Denote
(16)$$\begin{array}{c} B=\left(P\left(A_{i}▵A_{i}^{\prime }|C\right)<\sqrt{\epsilon })\;\mathrm{for}\;\mathrm{all}\;i\in \left\{1,\cdots ,I\right\}\right). \end{array}$$
From the Bonferroni inequality, we obtain $P\left(B^{c}\right)\leq I\sqrt{\epsilon }$. Subsequently, we observe that $|H(\alpha ||C)-H(\alpha^{\prime }\left||C\right)|\leq \log I$ holds almost surely. Hence,
(17)$$\begin{array}{cc} |H\left(\alpha |C\right)-H\left(\alpha^{\prime }|C\right)| & =|\int \left[H\left(\alpha |C\right)-H\left(\alpha^{\prime }|C\right)\right]dP|\\ & \leq P\left(B^{c}\right)\log I+\int_{B}\left|H(\alpha ||C)-H(\alpha^{\prime }\left||C\right)\right|dP\\ & \leq I\sqrt{\epsilon }\log I+\int_{B}\left|H(\alpha ||C)-H(\alpha^{\prime }\left||C\right)\right|dP. \end{array}$$
Function −xlogx is subadditive and increasing for $x\in (0,e^{-1}]$. In particular, we have |(x+y)log(x+y)−xlogx|≤−ylogy for x,y≥0. Thus, on the event *B* we obtain
(18)$$\begin{array}{cc} |H(\alpha ||C)-H(\alpha^{\prime }\left||C\right)| & =|\sum_{i=1}^{I}P\left(A_{i}^{\prime }|C\right)\log P\left(A_{i}^{\prime }|C\right)-\sum_{i=1}^{I}P\left(A_{i}|C\right)\log P\left(A_{i}|C\right)|\\ & \leq -\sum_{i=1}^{I}|P\left(A_{i}|C\right)-P\left(A_{i}^{\prime }|C\right)\left|\log\right|P\left(A_{i}|C\right)-P\left(A_{i}^{\prime }|C\right)|\\ & \leq -\sum_{i=1}^{I}P\left(A_{i}▵A_{i}^{\prime }|C\right)\log P\left(A_{i}▵A_{i}^{\prime }|C\right)\\ & \leq -I\sqrt{\epsilon }\log\sqrt{\epsilon } \end{array}$$
Plugging (Equation 18) into (Equation 17) yields the claim.  □

Now, we can prove the invariance of completion. Note that
(19)$$\begin{array}{c} I(\alpha ;\beta |C)=H(\alpha |C)+H(\beta |C)-H(\alpha \wedge \beta |C). \end{array}$$

**Proof** **of** **Theorem** **1.****1 (invariance of completion):** Consider some measurable fields A, B, and C. We are going to demonstrate
(20)$$\begin{array}{c} I(A;B|C)=I(A;\sigma (B)|C)=I(A;B|\sigma (C)). \end{array}$$Equality I(A;B|C)=I(A;B|σ(C)) is straightforward since P(A|C)=P(A|σ(C)) almost surely for all A∈J. It remains to prove I(A;B|C)=I(A;σ(B)|C). For this goal, it suffices to show that for any ϵ>0 and any finite partitions α⊂A and $\beta^{\prime }\subset \sigma \left(B\right)$ there exists a finite partition β⊂B such that
(21)$$\begin{array}{c} |I\left(\alpha ;\beta |C\right)-I\left(\alpha ;\beta^{\prime }|C\right)|<\epsilon . \end{array}$$Fix then some ϵ>0 and finite partitions $\alpha :={\left\{A_{i}\right\}}_{i=1}^{I}\subset A$ and $\beta^{\prime }:={\left\{B_{j}^{\prime }\right\}}_{j=1}^{J}\subset \sigma \left(B\right)$. Invoking Theorem 3, we know that for each η>0 there exists a class of sets ${\left\{C_{j}\right\}}_{j=1}^{J}\subset B$ which need not be a partition, such that
(22)$$\begin{array}{c} P(C_{j}▵B_{j}^{\prime })\leq \eta \end{array}$$
for all $j\in \left\{1,\cdots ,J\right\}$. Let us put $B_{J+1}^{\prime }:=\emptyset$ and let us construct sets $D_{0}:=\emptyset$ and $D_{j}:={⋃}_{k=1}^{j}C_{k}$ for $j\in \left\{1,\cdots ,J\right\}$. Subsequently, we put $B_{j}:=C_{j}\backslash D_{j-1}$ for $j\in \left\{1,\cdots ,J\right\}$ and $B_{J+1}:=\Omega \backslash D_{J}$. In this way, we obtain a partition $\beta :={\left\{B_{j}\right\}}_{j=1}^{J+1}\subset B$.The next step of the proof is showing an analogue of bound (Equation 22) for partitions β and $\beta^{\prime }$. To begin, for $j\in \left\{1,\cdots ,J\right\}$, we have
(23)$$\begin{array}{cc} P(B_{j}▵B_{j}^{\prime }) & =P\left(\left(C_{j}\backslash D_{j-1}\right)▵B_{j}^{\prime }\right)\leq P\left(C_{j}▵B_{j}^{\prime }\right)+P\left(D_{j-1}\cap B_{j}^{\prime }\right)\\ & \leq \eta +\sum_{k=1}^{j-1}P\left(C_{k}\cap B_{j}^{\prime }\right)\\ & \leq \eta +\sum_{k=1}^{j-1}\left[P\left(B_{k}^{\prime }\cap B_{j}^{\prime }\right)+P\left(\left(C_{k}\cap B_{j}^{\prime }\right)▵\left(B_{k}^{\prime }\cap B_{j}^{\prime }\right)\right)\right]\\ & \leq \eta +\sum_{k=1}^{j-1}\left[0+P(C_{k}▵B_{k}^{\prime })\right]\leq j\eta . \end{array}$$
Now, we observe for $j,k\in \left\{1,\cdots ,J\right\}$ and j≠k that
(24)$$\begin{array}{cc} P(C_{j}) & \geq P\left(B_{j}^{\prime }\right)-P\left(C_{j}▵B_{j}^{\prime }\right)\geq P\left(B_{j}^{\prime }\right)-\eta \\ P(C_{j}\cap C_{k}) & \leq P\left(B_{j}^{\prime }\cap B_{k}^{\prime }\right)+P\left(\left(C_{j}\cap C_{k}\right)▵\left(B_{j}^{\prime }\cap B_{k}^{\prime }\right)\right) \end{array}$$
(25)$$\begin{array}{cc} \; & \leq 0+P\left(C_{j}▵B_{j}^{\prime }\right)+P\left(C_{k}▵B_{k}^{\prime }\right)\leq 2\eta . \end{array}$$
Hence, by the Bonferroni inequality we derive
(26)$$\begin{array}{cc} P(B_{J+1}▵B_{J+1}^{\prime }) & =P\left(\left(\Omega \backslash D_{J}\right)▵\emptyset \right)=P\left(\Omega \backslash D_{J}\right)=1-P\left(D_{J}\right)\\ & \leq 1-\sum_{1\leq j\leq J}P\left(C_{j}\right)+\sum_{1\leq j<k\leq J}P\left(C_{j}\cap C_{k}\right)\\ & \leq 1-\sum_{1\leq j\leq J}P\left(B_{j}^{\prime }\right)+J\eta +\sum_{1\leq j<k\leq J}2\eta =J^{2}\eta . \end{array}$$Resuming our bounds, we obtain
(27)$$\begin{array}{cc} P(\left(A_{i}\cap B_{j}\right)▵\left(A_{i}\cap B_{j}^{\prime }\right)) & \leq P\left(B_{j}▵B_{j}^{\prime }\right)\leq J^{2}\eta \end{array}$$
for all $i\in \left\{1,\cdots ,I\right\}$ and $j\in \left\{1,\cdots ,J+1\right\}$. Then, invoking Theorem 4 yields
(28)$$\begin{array}{cc} |I\left(\alpha ;\beta |C\right)-I\left(\alpha ;\beta^{\prime }|C\right)| & \leq |H\left(\alpha \wedge \beta |C\right)-H\left(\alpha \wedge \beta^{\prime }|C\right)\left|+\right|H\left(\beta |C\right)-H\left(\beta^{\prime }|C\right)|\\ \; & \leq I\left(J+1\right)\sqrt{J^{2}\eta }\log\frac{I(J+1)}{\sqrt{J^{2}\eta }}+\left(J+1\right)\sqrt{J^{2}\eta }\log\frac{J+1}{\sqrt{J^{2}\eta }}. \end{array}$$
Taking η sufficiently small, we obtain (Equation 21), which is the desired claim.  □

Some consequence of the above result is this approximation result proved by Dobrushin [5] and Pinsker [6] and used by Wyner [7] to demonstrate the chain rule. Applying the invariance of completion, we supply a different proof than Dobrushin [5] and Pinsker [6].

**Theorem** **5**(split of join)**.**
*Let A, B, C, and D be subfields of J. We have*
(29)$$\begin{array}{c} I\left(A;B\wedge C|D\right)=\underset{\alpha \subset A,\beta \subset B,\gamma \subset C}{\sup}E\;I\left(\alpha ;\beta \wedge \gamma ||D\right), \end{array}$$
*where the supremum is taken over all finite subpartitions.*

**Proof.** Define class
(30)$$\begin{array}{c} E:=\underset{\beta \subset B,\gamma \subset C}{⋃}\sigma \left(\beta \wedge \gamma \right). \end{array}$$
It can be easily verified that E is a field such that σ(E)=σ(B∧C). Thus, for all finite partitions β⊂B and γ⊂C we have β∧γ⊂E. Moreover, by definition of E, for each finite partition ε⊂E there exists finite partitions β⊂B and γ⊂C such that partition β∧γ is finer than ε. Hence, by Theorem 2.4, we obtain in this case,
(31)$$\begin{array}{c} E\;I(\alpha ;\varepsilon ||D)\leq E\;I(\alpha ;\beta \wedge \gamma ||D)\leq I(\alpha ;E|D). \end{array}$$
In consequence, by Theorem 1.1, we obtain the claim
(32)$$\begin{array}{cc} I(A;B\wedge C|D) & =I\left(A;E|D\right)=\underset{\alpha \subset A,\varepsilon \subset E}{\sup}E\;I\left(\alpha ;\varepsilon ||D\right)\\ & =\underset{\alpha \subset A,\beta \subset B,\gamma \subset C}{\sup}E\;I\left(\alpha ;\beta \wedge \gamma ||D\right). \end{array}$$  □

The final approximation result which we need to prove the chain rule is as follows:
**Theorem** **6**(convergence of conditioning)**.**
*Let $\alpha ={\left\{A_{i}\right\}}_{i=1}^{I}$ be a finite partition and let C be a field. For each ϵ>0, there exists a finite partition $\gamma^{\prime }\subset \sigma \left(C\right)$ such that for any partition γ⊂σ(C) finer than $\gamma^{\prime }$ we have*
(33)$$\begin{array}{c} |H(\alpha |C)-H(\alpha |\gamma )|\leq \epsilon . \end{array}$$
**Proof.** Fix an ϵ>0. For each n∈N and A∈J, partition
(34)$$\begin{array}{c} \gamma_{A}:=\left\{\left((k-1)/n<P(A|C)\leq k/n\right):k\in \left\{0,1,\cdots ,n\right\}\right\} \end{array}$$
is finite and belongs to σ(C). If we consider partition $\gamma^{\prime }:={⋀}_{i=1}^{I}\gamma_{A_{i}}$, it remains finite and still satisfies $\gamma^{\prime }\subset \sigma \left(C\right)$. Let a partition γ⊂σ(C) be finer than $\gamma^{\prime }$. Then,
(35)$$\begin{array}{c} |P\left(A_{i}|C\right)-P\left(A_{i}|\gamma \right)|\leq 1/n \end{array}$$
almost surely for all $i\in \left\{1,\cdots ,I\right\}$. We also observe
(36)$$\begin{array}{cc} \left|H(\alpha |C)-H(\alpha |\gamma )\right| & \leq \int \left|H(\alpha ||C)-H(\alpha ||\gamma \right)|dP. \end{array}$$
We recall that function −xlogx is subadditive and increasing for $x\in (0,e^{-1}]$. In particular, we have |(x+y)log(x+y)−xlogx|≤−ylogy for x,y≥0. Hence, for n≥e we obtain almost surely
(37)$$\begin{array}{cc} \left|H(\alpha ||C)-H(\alpha ||\gamma )\right| & =|\sum_{i=1}^{I}P\left(A_{i}|C\right)\log P\left(A_{i}|C\right)-\sum_{i=1}^{I}P\left(A_{i}|\gamma \right)\log P\left(A_{i}|\gamma \right)|\\ & \leq -\sum_{i=1}^{I}|P\left(A_{i}|C\right)-P\left(A_{i}|\gamma \right)\left|\log\right|P\left(A_{i}|C\right)-P\left(A_{i}|\gamma \right)|\\ & \leq \frac{I\log n}{n}. \end{array}$$
Taking *n* so large that $n^{-1}I\log n\leq \epsilon$ yields the claim.  □

Taking the above into account, we can demonstrate the chain rule. Our proof essentially follows the ideas of Wyner [7], except for invoking Theorem 6.

**Proof** **of** **Theorem** **1.****2 (chain rule):** Let A, B, C, and D be arbitrary fields, and let α, β, γ, and δ be finite partitions. The point of our departure is the chain rule for finite partitions [9] (Equation 2.60)
(38)$$\begin{array}{c} I(\alpha ;\beta \wedge \gamma )=I(\alpha ;\beta )+I(\alpha ;\gamma |\beta ). \end{array}$$
By Definition 1 and Theorems 1.1, 5, and 6, conditional mutual information I(A;B|C) can be approximated by I(α;β|γ), where we take appropriate limits of refined finite partitions with a certain care.In particular, by Theorems 1.1, 5, and 6, taking sufficiently fine finite partitions of arbitrary fields B and C, the chain rule (Equation 38) for finite partitions implies
(39)$$\begin{array}{c} I(\alpha ;B\wedge C)=I(\alpha ;B)+I(\alpha ;C|B), \end{array}$$
where all expressions are finite. Hence, we also obtain
0=I(α;B∧C∧D)−I(α;D)−I(α;B∧C|D)=−I(α;B∧D)−I(α;D)−I(α;B|D)=−I(α;B∧C∧D)−I(α;B∧D)−I(α;C|B∧D)=I(α;B|D)+I(α;C|B∧D)−I(α;B∧C|D),
where all expressions are finite. Having established the above claim for a finite partition α, we generalize it to
(40)$$\begin{array}{c} I(A;B\wedge C|D)=I(A;B|D)+I(A;C|B\wedge D) \end{array}$$
for an arbitrary field A, taking its appropriately fine finite partitions.  □

## 3. Applications

This section borrows its statements largely from Dębowski [1,2,3] and is provided only to sketch some context for our research and justify its applicability to statistical language modeling. Let ${\left(X_{i}\right)}_{i\in Z}$ be a two-sided infinite stationary process over a countable alphabet X on a probability space $\left(X^{Z},X^{Z},P\right)$, where $X_{k}\left({\left(\omega_{i}\right)}_{i\in Z}\right):=\omega_{k}$. We denote random blocks $X_{j}^{k}:={\left(X_{i}\right)}_{j\leq i\leq k}$ and complete σ-fields $G_{j}^{k}:=\sigma \left(X_{j}^{k}\right)$ generated by them. By the generalized calculus of Shannon information measures, i.e., Theorems 1 and 2, we can define the entropy rate $h_{P}$ and the excess entropy $E_{P}$ of process ${\left(X_{i}\right)}_{i\in Z}$ as
(41)$$\begin{array}{cc} h_{P} & :=\lim_{n\to \infty }H_{P}\left(G_{0}|G_{-n}^{-1}\right)=H_{P}\left(G_{0}|G_{-\infty }^{-1}\right)\;\mathrm{if}\;X\;\mathrm{is}\;\mathrm{finite}, \end{array}$$
(42)$$\begin{array}{cc} E_{P} & :=\lim_{n\to \infty }I_{P}\left(G_{-n}^{-1};G_{0}^{n-1}\right)=I_{P}\left(G_{-\infty }^{-1};G_{0}^{\infty }\right), \end{array}$$
see [10] for more background.

Let $T\left({\left(\omega_{i}\right)}_{i\in Z}\right):={\left(\omega_{i+1}\right)}_{i\in Z}$ be the shift operation and let $I:=\left\{A\in X^{Z}:T^{-1}\left(A\right)=A\right\}$ be the invariant σ-field. By the Birkhoff ergodic theorem [11], we have $\sigma \left(I\right)\subset \sigma \left(G_{-\infty }\right)\cap \sigma \left(G_{\infty }\right)$ for the tail σ-fields $G_{-\infty }:={⋂}_{n=1}^{\infty }G_{-\infty }^{-n}$ and $G_{\infty }:={⋂}_{n=1}^{\infty }G_{n}^{\infty }$. Hence, by Theorems 1 and 2 we further obtain expressions
(43)$$\begin{array}{cc} h_{P} & =H_{P}\left(G_{0}|G_{-\infty }^{-1}\right)=H_{P}\left(G_{0}|G_{-\infty }^{-1}\wedge I\right)\;\mathrm{if}\;X\;\mathrm{is}\;\mathrm{finite}, \end{array}$$
(44)$$\begin{array}{cc} E_{P} & =I_{P}\left(G_{-\infty }^{-1};G_{0}^{\infty }\right)=H_{P}\left(I\right)+I_{P}\left(G_{-\infty }^{-1};G_{0}^{\infty }|I\right). \end{array}$$

Denoting the conditional probability F(A):=P(A|I), which is a random stationary ergodic measure by the ergodic decomposition theorem [12], we notice that $H_{P}\left(G_{0}|G_{-\infty }^{-1}\wedge I\right)=E{\;}_{P}H_{F}\left(G_{0}|G_{-\infty }^{-1}\right)$ and $I_{P}\left(G_{-\infty }^{-1};G_{0}^{\infty }|I\right)=E{\;}_{P}I_{F}\left(G_{-\infty }^{-1};G_{0}^{\infty }\right)$, and consequently we obtain the ergodic decomposition of the entropy rate and excess entropy, which reads
(45)$$\begin{array}{cc} h_{P} & =E{\;}_{P}h_{F}\;\mathrm{if}\;X\;\mathrm{is}\;\mathrm{finite}, \end{array}$$
(46)$$\begin{array}{cc} E_{P} & =H_{P}\left(I\right)+E{\;}_{P}E_{F}. \end{array}$$

Formulae (Equation 45) and (46) were derived by Gray and Davisson [13] and Dębowski [1] respectively. The ergodic decomposition of the entropy rate (Equation 45) states that a stationary process is asymptotically deterministic, i.e., $h_{P}=0$, if and only if almost all its ergodic components are asymptotically deterministic, i.e., $h_{F}=0$ almost surely. In contrast, the ergodic decomposition of the excess entropy (46) states that a stationary process is infinitary, i.e., $E_{P}=\infty$, if some of its ergodic components are infinitary, i.e., $E_{F}=\infty$ with a nonzero probability, or if $H_{P}\left(I\right)=\infty$, i.e., if the process is strongly nonergodic in particular, see [14,15].

The linguistic interpretation of the above results is as follows. There is a hypothesis by Hilberg [16] that the excess entropy of natural language is infinite. This hypothesis can be partly confirmed by the original estimates of conditional entropy by Shannon [17], by the power-law decay of the estimates of the entropy rate given by the PPM compression algorithm [18], by the approximately power-law growth of vocabulary called Heaps’ or Herdan’s law [2,3,19,20], and by some other experiments applying neural statistical language models [21,22]. In parallel, Dębowski [1,2,3] supposed that the very large excess entropy in natural language may be caused by the fact that texts in natural language describe some relatively slowly evolving and very complex reality. Indeed, it can be mathematically proved that if the abstract reality described by random texts is unchangeable and infinitely complex, then the resulting stochastic process is strongly nonergodic, i.e., $H_{P}\left(I\right)=\infty$ in particular [1,2,3]. Consequently, its excess entropy is infinite by formula (46). We suppose that a similar mechanism may work for natural language, see [23,24,25,26] for further examples of abstract stochastic mechanisms leading to infinitary processes.

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
