# Peer review of "Approximating Information Measures for Fields"

_entropy, 2020, doi:10.3390/e22010079_

Round 1

Reviewer 1 Report

This is a technically well written paper, which in my opinion is interesting for  some but not too many researchers. In order to attract a sufficent large number of readers, it would be nice to have more background justifying the results. In particular, relations to ergodic decomposition could be of some interest.

Author Response

As requested by Reviewer 1, I have added Section 3 Applications, in which I explain how the calculus of generalized Shannon information measures can be applied to the ergodic decomposition of stationary processes and statistical modeling of natural language. The added text cites 16 more references to provide the necessary background. This should motivate more clearly why the manuscript has been submitted to the special issue "Information Theory and language". Thank you for pointing out the need of such motivating section!

Reviewer 2 Report

Approximating Information Measures for Fields

In this work the author refers to a previous paper, published in 2009, where he presented a set of theorems regarding Shannon’s information measures. The author justifies this manuscript by indicating there are required fixes to the way Theorem 1 was presented in his paper of 2009.  The present manuscript exhibits an extremely rigorous treatment of the theme. No doubt this manuscript, together with those included as citations, will serve as a reference in the formal study of information theory. This work is well written, and I consider it valuable for publication in Entropy in its actual condition.

I think in the sentence (ine 49: “the approach proposed by Wyner [7] and Debowski [1] leads to a simpler and 49 more general calculus of Shannon information measures.”),  should be reviewed. Even if Wyner’s and Debowski’s approaches are equivalent, those are two approaches (separated by many years).  Another way to present this sentence might be adding a plural to the approach (es) by Wyner and Debowski.. Then it would read: the approaches proposed by Wyner [7] and Debowski [1] lead (with no ‘s’)  to a simpler…

Author Response

In response to Reviewer 2, I have corrected a bit my wordings throughout the manuscript. All changes have been highlighted in the red color. Although Reviewer 2 has not expressed such a request, in response to Reviewer 1, I have added Section 3 Applications in which I reiterate some applications of the generalized calculus of Shannon information measures to the ergodic decomposition and statistical modeling of natural language. I cite 16 more references.